Contributions to the functional morphology of caudate skulls: kinetic and akinetic forms

Natchev Nikolay 1 2 nikolay.natchev@univie.ac.at
Handschuh Stephan 3
Lukanov Simeon 4
Tzankov Nikolay 5
Naumov Borislav 4
Werneburg Ingmar 6 7 8 i.werneburg@gmail.com
1 Faculty of Natural Science, Shumen University , Shumen , Bulgaria
2 Department of Integrative Zoology, Vienna University , Vienna , Austria
3 VetCore Facility for Research, Veterinärmedizinische Universität Wien , Vienna , Austria
4 Institute of Biodiversity and Ecosystem Research, Bulgarian Academy of Sciences , Sofia , Bulgaria
5 Section Vertebrates, National Museum of Natural History, Bulgarian Academy of Sciences , Sofia , Bulgaria
6 Senckenberg Center for Human Evolution and Palaeoenvironment (HEP), Eberhard-Karls-Universität Tübingen , Tübingen , Germany
7 Fachbereich Geowissenschaften, Eberhard-Karls-Universität Tübingen , Tübingen , Germany
8 Museum für Naturkunde, Leibniz-Institute für Evolutions- and Biodiversitätsforschung, Humboldt Universität Berlin , Berlin , Germany
Abdala Virginia
Electronic publication date: 2016 Sep 20
Publication date: 2016
Volume: 4
Electronic Location ID: e2392
Received 2016 May 24; Accepted 2016 Aug 1
Copyright: © 2016 Natchev et al.
Copyright year: 2016
Copyright holder: Natchev et al.
License: This is an open access article distributed under the terms of the Creative Commons Attribution License, which permits unrestricted use, distribution, reproduction and adaptation in any medium and for any purpose provided that it is properly attributed. For attribution, the original author(s), title, publication source (PeerJ) and either DOI or URL of the article must be cited.
License URL: https://creativecommons.org/licenses/by/4.0/

Keywords: Feeding, Skull kinesis, μCT scanning, Urodela, Habitat shift, Prey shaking

Funding: Biology Department at Shumen University RD-08-66/02.02.2016 The study was partly supported by the Project number: RD-08-66/02.02.2016 of the Biology Department at Shumen University. We acknowledge support by Deutsche Forschungsgemeinschaft and Open Access Publishing Fund of University of Tübingen. We thank the “Fund for Support of the issue of publication in journals with Impact Factor (IF) and Impact Rang (SJR)” at Konstantin Preslavsky University Shumen. The funders had no role in study design, data collection and analysis, decision to publish, or preparation of the manuscript.

==============================
A strongly ossified and rigid skull roof, which prevents parietal kinesis, has been reported for the adults of all amphibian clades. Our μ-CT investigations revealed that the Buresch’s newt (Triturus ivanbureschi) possess a peculiar cranial construction. In addition to the typical amphibian pleurokinetic articulation between skull roof and palatoquadrate associated structures, we found flexible connections between nasals and frontals (prokinesis), vomer and parasphenoid (palatokinesis), and between frontals and parietals (mesokinesis). This is the first description of mesokinesis in urodelans. The construction of the skull in the Buresch’s newts also indicates the presence of an articulation between parietals and the exocipitals, discussed as a possible kind of metakinesis. The specific combination of pleuro-, pro-, meso-, palato-, and metakinetic skull articulations indicate to a new kind of kinetic systems unknown for urodelans to this date. We discuss the possible neotenic origin of the skull kinesis and pose the hypothesis that the kinesis in T. ivanbureschi increases the efficiency of fast jaw closure. For that, we compared the construction of the skull in T. ivanbureschi to the akinetic skull of the Common fire salamander Salamandra salamandra. We hypothesize that the design of the skull in the purely terrestrial living salamander shows a similar degree of intracranial mobility. However, this mobility is permitted by elasticity of some bones and not by true articulation between them. We comment on the possible relation between the skull construction and the form of prey shaking mechanism that the species apply to immobilize their victims.

Introduction

The crossopterygian ancestors of the tetrapods had neurokinetic skulls. However, early land vertebrate lineages lost this condition (Edgeworth, 1935; Laurin, 2010; Dutel et al., 2013; Dutel et al., 2015). The extreme forms of skull kinesis found in some lizards, snakes, and birds are judged to represent highly derived conditions (Iordansky, 1988; Evans, 2008).

The morphology of the skull in modern amphibians was studied in great detail (see Wake & Deban, 2000; Iordanskiĭ, 2000; Heiss et al., 2016). Whereas gymnophionans (see O’Reilly, 2000; Kleinteich et al., 2012) and anurans (see Iordansky, 1989) show only minute internal skull movement as adults, the degree of development of skull kinetics is rather different among urodeles (Marconi & Simonetta, 1988; Iordansky, 1988; Iordansky, 1989; Iordanskiĭ, 2000; Iordansky, 2001). Some species possess highly kinetic skulls during their larval stages to perform particular feeding modes (Iordansky, 1988). In the course of the ontogenetic development, however, the mobility between cranial bones substantially decreases (Stadtmüller, 1936; Iordansky, 1988; Iordansky, 1989; Marconi & Simonetta, 1988). Iordanskiĭ (2000) reported a high mobility in the rhinal section (prokinesis) of the skull in the Southern crested newt (Triturus karelinii), the Smooth newt (Lissotriton vulgaris), and the Common fire salamander (Salamandra salamandra). Information concerning the design of the joints which allow prokinetic movements in these species was lacking.

In the course of the present study, we investigated the cranial anatomy of a close relative of T. karelinii, namely the Balkan-Anatolian crested newt, T. ivanbureschi. Based on genetic data, this species was recently distinguished as a valid taxon from T. karelinii (Wielstra, Baird & Arntzen, 2013; Wielstra et al., 2013; Wielstra et al., 2014). Morphometric studies of the skull provided reliable information for specific identification based on cranial shape characteristics of both species (Ivanović et al., 2013; Ivanović & Arntzen, 2015).

After metamorphosis, S. salamandra is a terrestrial living species of similar body size compared to that of T. ivanbureschi (see Stojanov, Tzankov & Naumov, 2011). It belongs to the so called “true salamanders” (Salamandrinae), a group that is regarded as sister taxon to all newt species (Pleurodelinae) within Salamandridae (see Zhang et al., 2008; Pyron & Wiens, 2011). Some aspects of the feeding modes are rather different between newts and the “true salamanders.” The newts capture prey both on land, via tongue/jaw prehension (except for Pachytriton), as well as in water via suction feeding (except for the “basal” clade, containing Salamandrina). The “true salamanders” are adapted predominantly to terrestrial lifestyle and, as such, cannot perform suction feeding, which is reflected in their cranial morphology (Deban, 2003). According to Lukanov et al. (2016), the skeleton of the cranio-cervical complex is more fragile in T. ivanbureschi when compared to that of S. salamandra. The authors predicted that S. salamandra executes complex “shaking” or “killing” movements to subdue and immobilize their prey (sensu Dauth, 1983; Dauth, 1986; Natchev et al., 2015). In the range of the present study, we confirm that prediction and report on the specifics of the “prey shaking” behavior in S. salamandra. The execution of vigorous “prey shaking” on land requires strengthening of the skull and the cranio-cervical connection (see Natchev et al., 2015; Lukanov et al., 2016).

Using exactly the same scanning and 3D-reconstruction methods, we directly compare the architecture of the skull in T. ivanbureschi and S. salamandra and comment on obvious functional implications. On the base of our results, we discuss the potential impact of cranial kinesis on the execution of the prey capture and manipulation in newts. We provide the hypothesis that intracranial articulations in T. ivanbureschi affect the function of the jaws during the fast-closure-phase (see Bramble & Wake, 1985) of the feeding cycle (see Schwenk, 2000a).

Materials and Methods

The European distribution of T. ivanbureschi includes the south-eastern parts of the Balkan Peninsula with most of Bulgaria, the eastern parts of Greece, Macedonia and Serbia, as well as European Turkey. In Asia, its range encompasses the coastlines of the Aegean sea and the western and central coastlines of the Marmara sea up to 300 km inwards, but it is absent in the inner parts of Anatolia (Wielstra et al., 2014). Populations along the Asiatic Black sea coast, as well as the eastern coast of the Marmara sea, were very recently raised to species level and designated the name Triturus anatolicus, based on nuclear markers (Wielstra & Arntzen, 2016). In Bulgaria, T. ivanbureschi is ubiquitous across the country up to 1,700 m above sea level, but is absent around the Danube river and the lower parts of its tributaries (Stojanov, Tzankov & Naumov, 2011). It typically inhabits stagnant ponds with thick vegetation and their surroundings, and feeds on a variety ofinsect larvae, small crustaceans, earthworms, slugs, etc. T. ivanbureschi has well-defined terrestrial and aquatic stages and leaves the water during summer, after the breeding period. Most newts enter the water for hibernation during autumn, although some (mostly juveniles) spend the winter on land (Stojanov, Tzankov & Naumov, 2011). The specimens used in this study were caught in a pond near the village of Bistritsa in Sofia district (42.595184°N, 23.367833°E).

S. salamandra has a wide range from Western and Central Europe southwards to the south-eastern parts of the continent. In Bulgaria, it is ubiquitous in the mountainous parts of the country (up to 2,350 m above sea level, usually between 800 and 1,600 m) and is absent in Strandzha, the Black sea coast, and most of the Danube valley. It lives on land, prefers humid forested areas and is usually nocturnal. It feeds on earthworms, slugs, arthropods and their larvae (Stojanov, Tzankov & Naumov, 2011). The adult specimens used in this study were caught near the village of Bov in Sofia district (43.016351°N, 23.367623°E).

Osteology of the skull was investigated using x-ray micro-computed tomography (μCT). Two adult specimens from both investigated species were provided from the collection of the National Museum of Natural History of Bulgaria (NMNH, Sofia). The animals were fixed in 4% formaldehyde, washed, and preserved in 70% ethanol. They were mounted in 70% ethanol and scanned for bone structures using a μCT35 (SCANCO Medical AG, Brüttisellen, Switzerland) with 70 kV source voltage and 114 μA intensity. The reconstructed images (Figs. 1 and 2) were visualized via volume rendering using Drishti (Limaye, 2012). Nomenclature of skeletal elements follows Iordansky (1989).

Figure 1 Craniocervical osteology of Triturus ivanbureschi based on μCT-reconstruction.

(A) Lateral view with (B) sagittal section, and (C) dorsal view. Cranial joints are indicated. Legend: a, atlas; bb, basibranchial; ch, ceratohyal; cb-I, ceratobranchial I; d, dentary; eb-I, epibranchial I; eo, exoccipital; f, frontal; m, maxilla; n, nasal; os, orbitosphenoid; sq, squamosum; pf, prefrontal; pm, premaxilla; ps, parasphenoid; pq, palatoquadrate; pt, pterygoid; q, quadrate, v, vomer.

Figure 2 Craniocervical osteology in Salamandra salamandra based on μCT-reconstruction.

(A) Lateral view with (B) sagittal section, and (C) dorsal view. For abbreviations see Fig. 1.

To investigate the specifics of prey manipulation in S. salamandra, we filmed two living specimens each in lateral view when feeding on Tenebrio sp. larvae (Figs. 3E–3H). This study was in compliance with the national laws of Bulgaria (collection-permit No. 520/23.04.2013) and the international requirements for ethical attitude towards animals. The animals were housed in the zoological laboratory at Vienna University. We produced three films per specimen using the digital high-speed camera system Photron Fastcam-X 1024 PCI (Photron limited; Tokyo, Japan) at 1,000 fps. We used a highly light-sensitive objective AF Zoom—Nikkor 24–85 mm (f/2,8-4D IF). Two “Dedocool Coolh” tungsten light heads with 2 × 250 W (ELC), supplied by a “Dedocool COOLT3” transformer control unit (Dedo Weigert Film GmbH; München, Germany). Feeding behaviour was compared to that of T. ivanbureschi (Figs. 3A–3D; Lukanov et al., 2016).

Figure 3 Prey shaking behavior in both investigated species.

Comparison of prey manipulation in Triturus ivanbureschi (A–D) and Salamandra salamandra (E–H) using selected frames from video sequence shots at 420 fps (T. ivanbureschi) and 1,000 fps (S. salamandra). Both species have prey in their mouth and the mouth is so far closed in both species. Whereas T. ivanbureschi shows single side movements of the body during prey shaking on land, S. salamandra shows a complex prey shaking behavior.

Results

As typical for urodelans (see Marconi & Simonetta, 1988), the μCT-reconstructions revealed a primarily flat skull roof composed of flat pairs of frontal and parietal bones in both investigated species. In T. ivanbureschi, the joint between the squamosal, as a palatoquadrate associated structure, and the parietal (Fig. 1) permits the typical amphibian pleurokinetics (see Iordanskiĭ, 2000). In S. salamandra, these bones are firmly attached (Fig. 2). In general, in the skull of S. salamandra, the major bone elements are tightly sutured with each other suggesting a very limited (if any) potential for skull kinesis (Fig. 2).

In T. ivanbureschi, the nasals and the frontals are not firmly fused and there is a detectable gap between these elements (see Figs. 1B and 1C). These articulations appear to permit prokinesis in this species. The contralateral calvarian plates overlap each other at the median line of the skull, but do not suture firmly (Figs. 1B and 1C). In addition, parietals and frontals are not firmly attached to each other and the sagittal section through the skull reveals a gap between the bones (Fig. 1B), which appears to permit mesokinetic movements. The construction of the joint between the exoccipital and the parietal is stronger fused than between the other mentioned elements, but it is not fixed as in S. salamandra, and a gap is visible between these bones (Fig. 1C).

In T. ivanbureschi, prey shaking on land is represented in single side movements of the body (Figs. 3A–3D; Lukanov et al., 2016). In S. salamandra, however, the prey shaking movements occurred in grouped clusters including up to five shakes. The prey shake modus in S. salamandra is a complex behavior including series of ventrolateral flexions of the whole body (Figs. 3E–3H), including the tail. This way, S. salamandra is hitting and dragging the prey against the substrate allegedly inflicting severe damages on the victim.

Discussion

Our analyses revealed a unique skull construction in T. ivanbureschi consisting of flexible articulations in the skull unknown to adult urodeles to this date. Compared to T. ivanbureschi, the more “typical” caudate S. salamandra has a strongly sutured skull permitting only little intracranial articulation. We suppose, however, that the net degree of mobility of certain skull segments is similar in both species, because the kinetic restrictions of tight sutures in S. salamandra will be neglectable due to the high elasticity of the bones. This is possible because the skull roof and palatal bones of S. salamandra are much thinner compared to T. ivanbureschi. In other words, the disadvantage of having thicker bones in T. ivanbureschi, namely less elasticity, is circumvented by the presence of kinetic intracranial joints.

The skull kinesis in adult salamandrids is usually reduced to pleuro- and prokinesis. Iordansky (1982) and Iordansky (1989) defined prokinesis as elastic attachment between the nasals and the prefrontals and the prefrontals and the frontals. This kind of skull kinesis is typical for hinobiids and ambystomids, but is rare in salamandrids. Prokinesis may occur only passive because urodeles lack specialized rhinal muscles (Iordansky, 1989; Iordanskiĭ, 2000). In T. ivanbureschi, the construction of the joint between the nasals and the frontals indicates that the preorbital section of the skull roof can rotate against the posterior cranial sections when certain elastic forces are applied. Such prokinetic junction was reported earlier for T. karelinii (Iordanskiĭ, 2000).

In the skulls of T. ivanbureschi, in addition to prokinesis and pleurokinesis, we discovered meso- and palatokinetic articulations (see Fig. 1). High speed films of feeding T. ivanbureschi specimens revealed that the head actually shows intracranial deformation visible in external view (N. Natchev, 2016, personal observations; Lukanov et al., 2016).

Intracranial movements are known for some salamander larvae (see Deban & Marks, 2002). However, cranial kinesis is largely reduced in postmetamorphic salamanders (Stadtmüller, 1936; Iordansky, 1988; Iordansky, 1989; Marconi & Simonetta, 1988). According to Stadtmüller (1936), during the ontogeny in some urodelans, the cranial kinesis is usually lost. The skull is highly kinetic in young larvae. Afterwards, the skull becomes akinetic in older larvae, and kinetic skulls may secondarily reappear in adults. However, the simplest explanation for the origin of the skull roof kinetics in T. ivanbureschi would be that pro-, palato-, meso-, and eventually metakinesis are present in the larval stages and the elastic connections are retained into adulthood. As such, we propose that the skull kinesis in T. ivanbureschi might represent a neotenic feature. However, the ontogenetic development of the skull has to be investigated in detail before providing a final judgement.

The skull morphology of the close relative of T. ivanbureschi, T. karelinii, was studied by Iordansky (1988) and Iordanskiĭ (2000) using classical dissections and mechanical manipulations. No mesokinetic joint was found by the author. Until recently, the whole population of T. ivanbureschi was considered to belong to T. karelinii (see Wielstra, Baird & Arntzen, 2013; Wielstra et al., 2013; Wielstra et al., 2014). It is possible that T. karelinii simply lacks a mesokinetic articulation, but it is also possible that the specimen dissected by Iordansky (1988) was fixed in a manner to harden the joints between the frontals and the parietals. Further investigations on the skull osteology in newts from different localities will provide data explaining whether mesokinesis is typical only for T. ivanbureschi or whether it can be found in local groups of T. karelinii, or in the newly described T. anatolicus (Wielstra & Arntzen, 2016).

The specialized head morphology of T. ivanbureschi (Fig. 1) indicates to a type of skull kinesis, which is, in some regards, analogous to that of mesokinesis found in some extant lizards (Frazzetta, 1962; Frazzetta, 1983; Frazzetta, 1986; Bramble & Wake, 1985; Iordansky, 1989; Schwenk, 2000b; Payne, Holliday & Vickaryous, 2011; Mezzasalma, Maio & Guarino, 2014; Montuelle & Williams, 2015). Despite not being able to provide irrefutable evidences for metakinesis, we propose a possible kinetic connection between the exoccipitals and the parietals in T. ivanbureschi. Kinetic junction between cranial bones can be predicted on the base of the morphology of the joints connecting these bones (see Mezzasalma, Maio & Guarino, 2014). In T. ivanbureschi we found, that there is a visible gap in the joints between the exoccipital and the parietal bones (see Fig. 1B). The design of this construction indicates on possible mobility in these connections.

However, even if metakinesis is present, we cannot consider an urodelan skull as amphikinetic in the sense used for lizards. We propose that the convergent development of meso- (and perhaps meta-) kinetic joints in lizards and in newts represents an adaptation to different aspects of feeding performance.

In lizards, the amphikinetic mobility serves to increase the range of longitudinal movements of the palate-maxillary complex in relation to the skull roof (Herrel et al., 1999; Iordanskiĭ, 2000; Mezzasalma, Maio & Guarino, 2014). This permits an “active kinesis” by which the jaws better fit to the grasped food item and the myovectors of the jaw muscles work more effectively. The overall result is an improvement of the prey holding and manipulation ability of the jaws (see Bramble & Wake, 1985; Iordanskiĭ, 2000; Schwenk, 2000b; Montuelle & Williams, 2015). In T. ivanbureschi, the mechanism is rather different (Lukanov et al., 2016) and might be based on a combination of pro-, pleuro-, palatal-, meso-, and potentially metakinesis.

T. ivanbureschi uses hydrodynamic based mechanisms in underwater feeding (Lukanov et al., 2016). Aquatic feeding newts benefit from pleurokinesis because it allows for reaching a broader gape (Iordanskiĭ, 2000; Iordansky, 2001; Deban & Wake, 2000). When feeding on smaller and/or more elusive prey, the increased velocity of the jaw closing during the fast closing phase (see Bramble & Wake, 1985) will be of an advantage for a predator. Our functional analysis indicates (Table 1) that the lack of rigid “skull table” in T. ivanbureschi may benefit the jaw closing mechanism. Compared to other adult newts studied so far (see Heiss, Aerts & Van Wassenbergh, 2013; Heiss, Aerts & Van Wassenbergh, 2015), T. ivanbureschi needs significantly less time for jaw closure (see Lukanov et al., 2016 and Table 1). A detailed investigation of the skull roof morphology in the Danube crested newt (T. dobrogicus), the Alpine newt (Ichtyosauraalpestris), and the Smooth newt (Lissotriton vulgaris) indicated that the skulls in these newts lack the elastic connections between the frontals and the parietals and between the parietals and the exoccipitals (see Kucera, 2013; Heiss et al., 2016). We propose that the jaw closing mechanism in T. ivanbureschi is supported by elastic forces that permit faster execution of the movements. During jaw opening, the epaxial musculature contracts and rotates the parietals in dorsal abduction (metakinesis) and the frontals move passively (mesokinesis). During jaw opening, the flat bony elements of the longitudinal axis of the mouth roof (Fig. 1) would be mechanically loaded by the activation of the powerful dorsalis trunci muscles. When jaw closure starts, the potential elastic energy stored in the bony elements of the mouth’s dorsal axis would be released and contributes for faster jaw closure and potentially for deeper penetration of the teeth into the prey, as found in small-sized lizards (see Montuelle & Williams, 2015).

Table 1 Comparison of the duration of the jaw closing phase.

Difference between means of duration of fast jaw closure cycle during aquatic and terrestrial feeding among different salamandrid species.

Species	Life stage	Medium	N	Mean (ms)	SD	p	Source	
Lissothriton vulgaris	Aquatic	Water	20	21	± 4	0.193	Heiss, Aerts & Van Wassenbergh (2015)	
Terrestrial		5	28	± 3	0.001		
Aquatic	Air	20	40	± 11	< 0.0001		
Terrestrial		20	34	± 9	< 0.0001		
Ichthyosaura alpestris	Aquatic	Water	20	27.1	± 4.9	< 0.0001	Heiss, Aerts & Van Wassenbergh (2013)	
Terrestrial		20	32	± 5.4	< 0.0001		
Aquatic	Air	20	32.6	± 7.8	< 0.0001		
Terrestrial		20	32.8	± 8.5	< 0.0001		
Triturus ivanbureschi	Terrestrial	Water	49	19.19	± 5.63	–	Lukanov et al. (2016)	
	Air	60	13.34	± 4.22			

The elastic connection between some elements of the skull may impact negatively the stability of the whole construction. According to Lukanov et al. (2016), T. ivanbureschi shows a rather simple prey shaking mode on land (Figs. 3A–3D). The shake cycles are isolated and not grouped in clusters. T. ivanbureschi uses simple, strictly horizontal lateral abduction of the body during the prey shaking. In the predominantly terrestrial living salamandrid S. salamandra, the skull is almost akinetic (i.e., no true kinetic joints) and the only moving element is the lower jaw. This design is consistent with the prey manipulation behavior of the species. In contrast to the semi-aquatic T. ivanbureschi, S. salamandra uses successive ventro-lateral flexions of the body and the neck during prey shaking (Fig. 3). The predator hits and drags the prey against the substrate in rapid and repetitive left-right alternations. This type of prey manipulation demands a rigid skull frame and the kind of skull kinesis found herein would weaken the construction. The fact that the bones of S. salamandra show a greater elasticity when compared to T. ivanbureschi does not weaken this argument. The strongly sutured skull of S. salamandra permits internal stability during prey shaking, but for the subsequent feeding cycle, the skull is elastic enough to maintain strong biting forces.

Our results are in line with the previously published data concerning the skull osteology in S. salamandra (for overview see Francis, 1934), which possesses a typical amphibian non-flexible skull roof with fully fused sutures in adult stage. According to Iordanskiĭ (2000), the skull in S. salamandra is rhynchokinetic (i.e., the snout tip is moveable against the rest of the skull). Our data indicate non-elastic connections between the nasals, prefrontals, and frontals (see Fig. 2). The bones which build the nasal region are flat and rather thin. It is possible that the mechanical manipulation performed by Iordanskiĭ (2000) had induced mechanical bending of the bones and no (or only minimal) kinetic movement is allowed by the joints. In S. salamandra, the squamosum is firmly attached to the exoccipital and pleurokinetic movements are rather constrained (Fig. 2).

In conclusion, we have to note that in addition to histology the μCT scanning technique is a powerful tool for studying cranial articulations in small-sized vertebrates. In many salamandrids, for example, most of the bones building the skull are flat and thin. Such bones can be easily bent and loaded during pincette manipulations (as used e.g. by Marconi & Simonetta, 1988; Iordansky, 1988; Iordanskiĭ, 2000). The mechanical bending and movement of defined skull segments against each other do not permit precise identification of the kinetic joints. As such, for precise analysis, the cranial joints have to be investigated on micromorphological level and need to be related to functional requirements.

We thank to Josef Weisgram and Martin Glösmann for material support. Madelaine Böhme is thanked for discussion.

Additional Information and Declarations

Competing Interests

Author Contributions

Animal Ethics

Field Study Permissions

Data Deposition

The authors declare that they have no competing interests.

Nikolay Natchev conceived and designed the experiments, performed the experiments, analyzed the data, wrote the paper, prepared figures and/or tables, reviewed drafts of the paper.

Stephan Handschuh conceived and designed the experiments, performed the experiments, contributed reagents/materials/analysis tools, prepared figures and/or tables, reviewed drafts of the paper.

Simeon Lukanov performed the experiments, contributed reagents/materials/analysis tools, prepared figures and/or tables, reviewed drafts of the paper.

Nikolay Tzankov analyzed the data, contributed reagents/materials/analysis tools, reviewed drafts of the paper.

Borislav Naumov contributed reagents/materials/analysis tools, reviewed drafts of the paper.

Ingmar Werneburg conceived and designed the experiments, analyzed the data, wrote the paper, prepared figures and/or tables, reviewed drafts of the paper.

The following information was supplied relating to ethical approvals (i.e., approving body and any reference numbers):

This study was in compliance with the national laws of Bulgaria (collection-permit No. 520/23.04.2013).

A special ethical approval commission does not exists and according to our legislation (Biodiversity Act), permits are issued by the Ministry of Environment and Waters (MoEW) for the scientists that are allowed to handle and collect vertebrates. The animals were not sacrificed for the purposes of the study, but were provided as fixed specimens from the collection of the Museum of Natural History (Sofia, Bulgaria) from Dozent Nikolay Tzankov.

The following information was supplied relating to field study approvals (i.e., approving body and any reference numbers):

This study was in compliance with the national laws of Bulgaria (collection-permit No. 520/23.04.2013).

The following information was supplied regarding data availability:

Our raw data are represented in the figures and tables. Table 1 represents data from previously published studies.

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
