# Peer review of "Contributions to the functional morphology of caudate skulls: kinetic and akinetic forms"

_PeerJ, doi:10.7717/peerj.2392_

## Round 0.1 · original submission · Major Revisions

I have received two reviews of your manuscript. Both reviewers agreed that it is well written and illustrated. One shared concern by both reviewers refers to the fixing influence on the mechanical properties of the tissues. This aspect concerns me also, so a convincing explanation should be giving here. Another issues in your experimental design have been questioned too, and this is a main point considering the objectives of your study. Descriptive data seem to be the most significant contribution of your work, and I welcome them because a great deal of row data are still needed in the framework of the vertebrate comparative morphology. However, I concur with our reviewers in that methodological flaws make your inferences about mechanical aspect of the caudate skull problematic. Thus, as it stands the manuscripts should focus on the morphological descriptions. Otherwise, your experimental work should be more convincingly performed.

Reviewer 1 ·

Basic reporting

Dear Editor,

I have considered now the paper by Nikolay Natchev et al. on ‘Contributions to the functional morphology of caudate skulls: kinetic and akinetic forms’. This paper gives a morphological, postural/kinematical and behavioural description/comparison of the skulls and feeding behaviour of a species with a kinetic skull (Triturus ivanbureschi) and a species with a premised akinetic skull (Salamandra salamandra).
The manuscript is well written and illustrated. The topic and research questions are sufficiently introduced and results are adequately discussed (see further).

Experimental design

Sample sizes are small (but see further)

questions on methods

It is mentioned that individuals were caught: these were only used for the behavioural study ??

The Xray-measurements are on fixed animals….does this concern museum specimens or freshly killed and fixed specimens? Fixing might influence tissue properties.

For the manipulation experiments, forces in all directions were applied. How do these direction compare to these of in vivo forces (during prey manipulation)? What about the magnitude of these forces compared to expected in vivo forces ? Ultimately everything will move (or bend).

About the elastic deformation: might fixation have resulted in decalcification (hence unrealistic deformations/movements).

On line 139: What is exactly meant with ‘on the perpendicular’?

Validity of the findings

Despite the fact that the sample sizes are small, the results seem reliable and are discussed with sufficient reserve and conclusions are formulated conditionally. This paper is largely descriptive and the observations seem (I am not entirely familiar with the topic) to be framed in the current knowledge on skull kinesis in amphibians (and beyond). As such I have few comments to make.

Reviewer 2 ·

Basic reporting

The manuscript appears to follow the general requirements of PeerJ. There are some minor grammatical errors, but these can easily be fixed. The figures are relevant and of good quality and enhance the presentation of the work.

Experimental design

The most significant problem with this manuscript is in describing certain aspects of the experimental design. As presented, some methods arguably could not be replicated by other investigators. Particularly problematic is the procedure involving the measurement of angles formed by various parts of the skull when external forces are applied. For example: How was the skull firmly fixed in place by putting the neck into a clamp? Did this somehow completely immobilize the skull so that internal measurements of displacement could be made? This is difficult to picture. What kind of instrument was used to apply forces to the skull? Was this done by hand or using some custom-built apparatus? What were the forces applied to different parts of the skull? How were force magnitudes and vectors were kept consistent? Knowing the exact forces applied, as well as the vectors with which they were applied, is critical to the procedures being repeatable by other investigators (for example by those who want to examine other species and make comparisons to this study).

Another significant issue, which may warrant the complete removal of the skull manipulation experiments, is that they specimens used were fixed. Formalin/formaldehyde fixation greatly affects the mechanical properties of tissues. For example, bones become softer and more flexible, yet brittle. I would hesitate to make any conclusions about bone flexibility, joint mobility, etc. based on the manipulation of fixed specimens.

Validity of the findings

The authors admit that they cannot say much statistically due to their small sample size, and the difficulty of acquiring specimens for study is noted. However, as discussed above, we cannot make much of these measurements (regardless of how many specimens were used) if we do not know the details of how external forces were applied. One question is why do the authors bold the largest angles that were induced? It seems that because different sets of bones/joints are involved in each manipulation, that comparisons of raw angle changes would not be comparable among them. Between species, perhaps comparisons could be made, but not among sets of structures. As indicated above, the fact that preserved specimens were used is at least as significant in terms of accurately estimating possible in vivo biomechanics.

I suggest focusing more on the imaging and morphological description, and not including the manipulations, which clearly are problematic.

---

## Round 0.2 · Minor Revisions

Many thanks for your carefully consideration of our suggestions. I am ready to accept your paper, but a few minor points require your further attention. I do not quite understand your Fig. 3. Why is a close mouth in 3B compared with an opened one? It would be more logic to compare the opened mouth in both species.

Please, define rhynchokinetic (page 13).

---

## Round 0.3 · accepted · Accept

Thank you for adding what I requested. I think that the manuscript is ready to be published.